# Development of a High Throughput Screen for the Identification of Inhibitors of Peptidoglycan *O-*Acetyltransferases, New Potential Antibacterial Targets

**DOI:** 10.3390/antibiotics8020065

**Published:** 2019-05-27

**Authors:** Ashley S. Brott, Carys S. Jones, Anthony J. Clarke

**Affiliations:** Department of Molecular and Cellular Biology, University of Guelph, Guelph, ON N1G 2W1, Canada; abrott@uoguelph.ca (A.S.B.); carys@uoguelph.ca (C.S.J.)

**Keywords:** peptidoglycan, *O-*acetyltransferase, antibiotic target, high throughput screening, HTS

## Abstract

The *O*-acetylation of peptidoglycan occurs in many Gram-negative and most Gram-positive pathogens and this modification to the essential wall polymer controls the lytic activity of the autolysins, particularly the lytic transglycosylases, and inhibits that of the lysozymes of innate immunity systems. As such, the peptidoglycan *O*-acetyltransferases PatA/B and OatA are recognized as virulence factors. In this study, we present the high throughput screening of small compound libraries to identify the first known inhibitors of these enzymes. The fluorometric screening assay developed involved monitoring the respective *O*-acetyltransferases as esterases using 4-methylumbelliferylacetate as substrate. Pilot screens of 3921 compounds validated the usefulness of the HTS protocol. A number of potential inhibitors were identified amongst a total of 145,000 low molecular-weight compounds, some of which were common to both enzymes, while others were unique to each. After eliminating a number of false positives in secondary screens, dose response curves confirmed the apparent specificity of a benzothiazolyl-pyrazolo-pyridine as an inhibitor of *Neisseria gonorrhoe*ae PatB, and several coumarin-based compounds as inhibitors of both this PatB and OatA from *Staphylococcus aureus*. The benzothiazolyl-pyrazolo-pyridine was determined to be a non-competitive inhibitor of PatB with a *K*_i_ of 126 µM. At 177 µg/mL and close to its solubility limit, this compound caused a 90% reduction in growth of *N. gonorrhoeae*, while growth of *Escherichia coli*, a bacterium that lacks PatB and, hence, does not produce O-acetylated peptidoglycan, was unaffected. These data provide preliminary proof of concept that peptidoglycan *O*-acetyltransferases would serve as useful antibacterial targets.

## 1. Introduction

Given its uniqueness to bacteria, together with its essential role in maintaining cell viability, the metabolism of peptidoglycan (PG) has been exploited, both naturally and clinically, as a prime target for antibacterial action. Thus, in addition to the clinical use of antibiotics that block PG biosynthesis, such as the β-lactams and vancomycin, the hydrolysis of PG represents the first line of defense against invading pathogens by the innate immune system. Specifically, lysozymes present in blood, the liver, in secretions such as milk, tears, urine, and saliva, at mucosal surfaces, and in phagocytes, including macrophages, neutrophils, and dendritic cells [1], act to hydrolyze the β-(1→4) linkage between the repeating *N*-acetylmuramoyl (MurNAc) and *N*-acetylglucosaminyl (GlcNAc) residues that constitute the glycan chains of PG. Such lysis leads to the rapid rupture of the invading bacteria, while the released PG fragments act as critical activators for further immune response [2,3]. This defensive action of lysozyme is effective against both Gram-positive and Gram-negative bacteria, despite the fact that the PG sacculus of the latter is covered by their outer membrane. Other factors of the innate immune system, such as defensins [4] and lactoferrin [5], serve to disrupt the outer membrane to facilitate exposure of PG to lysozyme. However, many pathogens protect themselves from lysozyme by chemically modifying their PG through O-acetylation. 

PG O-acetylation occurs at the C-6 hydroxyl group of MurNAc residues within the sacculus. This modification sterically inhibits the productive binding of lysozyme [6] in a concentration dependent manner [7,8,9,10]. Both Gram-positive and Gram-negative bacteria produce *O*-acetyl-PG, but it appears to be predominant in pathogens. For example, only pathogenic species of *Staphylococcus* (including *S. aureus*) produce O-acetylated PG, and each is highly resistant to lysozyme. In contrast, all non-pathogenic species of the genus both lack this modification and they are lysozyme sensitive [11]. As a non-stoichiometric modification, the extent of PG O-acetylation varies with species and strain (typically ranging between 20% and 70% [12,13,14,15]) with the age of a bacterial culture [16] and differential state (e.g., vegetative cells vs. swarmers [17], or viable but not culturable [16]). The increased susceptibility of PG with decreased levels of O-acetylation to host lysozyme correlates directly with the decrease in pathogenicity of both Gram-positive (*S. aureus* [18,19], *S. suis* [20], *Streptococcus iniae* [21], *E. faecalis* [22], *Listeria monocytogenes* [23,24,25]*, Bacillus subtilis* [26]) and Gram-negative (*Helicobacter pylori* [27], *Neisseria gonorrhoeae* [10,28] and *N. meningitidis* [29]) pathogens. In all cases, either the enzyme directly responsible for PG O-acetylation and/or its regulator(s) have been identified as a critical virulence factor [10,18,19,20,21,22,23,24,25,26,27,28,29]. 

In addition to providing increased resistance to lysozyme, PG O-acetylation increases resistance to β-lactam antibiotics [30] and controls endogenous autolytic activity associated with PG metabolism [14,31,32,33,34,35], including that required for cell septation [32,33]. In Gram-negative bacteria, the lytic transglycosylases (LTs) represent a major class of autolysins involved with PG metabolism and cell septation [36]. These enzymes lyse PG at the same site as the lysozymes, but instead of hydrolysis, they utilize the C-6 hydroxyl group of MurNAc to generate 1,6-anhydroMurNAc products [37]. Consequently, the presence of a C-6 *O*-acetyl group precludes LT activity and, thus, provides a means of autolytic control at the substrate level [14,31,36].

The O-acetylation of PG is a maturation event, occurring after transglycosylation of Lipid II precursors into the sacculus [15,38], which requires that the enzymes catalyzing the modification are extracytoplasmic. As such, these enzymes present attractive targets for the development of a new class of antibacterials. Inhibitors would be expected to (i) act as anti-virulence factors to render pathogens more susceptible to the host immune system; (ii) possibly act synergistically with existing antibiotics that depend on autolysins for their bactericidal effect, especially the β-lactams; (iii) potentially act directly as bactericidal agents by removing LT control thereby permitting rampant and catastrophic autolysis; (iv) be less susceptible to efflux as the targets are either periplasmic or external in Gram-negative and Gram-positive pathogens, respectively; and (v) be ineffective against the predominantly non-O-acetylated commensal microbiota. 

In Gram-negative bacteria, PG O-acetylation is catalyzed by the concerted action of two enzymes, PG *O*-acetyltransferase (Pat) A and B [39,40]. As a putative membrane-bound *O*-acyl transferase (MBOAT), PatA is predicted to function as a transmembrane shuttle of acetyl groups from cytoplasmic pools to the periplasmic face of the inner membrane [12,13,14] (Figure 1). The translocated acetyl is then presented to PatB for its transfer to the C-6 hydroxyl groups of MurNAc residues in the existing PG sacculus. With Gram-positive bacteria, a single bimodal integral membrane protein, PG *O*-acetyltransferase A (OatA), catalyzes both reactions [18,19,41]. With its 11 predicted transmembrane helices [42], the N-terminal domain of OatA (OatA_N_) is thought to function like PatA for the translocation of acetyl groups [12,13], while the extracellular C-terminal domain (OatA_C_) catalyzes their transfer to muramoyl residues [41] (Figure 1). Both recombinant PatB and OatA_C_ have been biochemically and kinetically characterized [39,41,43,44,45], and the X-ray crystal structure of OatA_C_ has been determined at high resolution [41]. They are members of the SGNH/GDSL hydrolase superfamily (CCD cl01053) and, thus, they possess a canonical Ser-His-Asp catalytic triad that catalyzes the *O*-acetyltransfer reaction via a ping-pong bisubstrate-biproduct mechanism of action [41,45]. 

With the ever-increasing incidence of antibiotic resistance amongst pathogens that produce O-acetylated PG, and which have been identified by the World Health Organization as high-level priority pathogens in need of research and development of new antibiotics, we sought to identify inhibitors that could be used to prove the concept that PatB and/or OatA_C_ represent novel antibacterial targets. Herein, we present the development of a high throughput screen and its application for the identification of the first known inhibitors of both *N. gonorrhoeae* PatB (*Ng*PatB) and *S. aureus* OatA_C_ (*Sa*OatA_C_). 

## 2. Results

### 2.1. Production, Purification and Stability of NgPatB and SaOatA_C_ Constructs 

Soluble constructs of *Sa*OatA_C_ [41] and *Ng*PatB [39,44,45] have been used in previous studies to characterize PG *O-*acetyltransferase activity in vitro. Both *Sa*OatA_C_ and *Ng*PatB_Δ69_, previously referred to as *Ng*PatB_Δ77_, catalyze weak esterase activity toward fluorogenic and chromogenic artificial substrates [39,41,46], and they are stable for the time frame needed to complete a pilot screen for potential inhibitors [39,41]. As the addition of an N-terminal SUMO-tag had been shown previously to stabilize other constructs of *Ng*PatB [39], we engineered a SUMO-*Ng*PatB_Δ69_ fusion protein to aid in the production of large quantities of recombinant *Ng*PatB_Δ69_. Whereas stability was qmaintained throughout the time course of the pilot screen (described below), we observed that over longer durations of time, *Ng*PatB_Δ69_ experienced partial degradation while retaining full specific activity. The degradation, as visualized by SDS PAGE (data not shown), appeared to be consistent, suggesting that a stable active core of the enzyme existed. Given the extended time frame associated with a large high throughput screen (HTS), the availability and use of a construct representing this stable core would be more appropriate. Using liquid chromatography quadrupole-time-of-flight mass spectrometric (LC-Q-TOF MS) analysis, we determined that the stable *Ng*PatB core had a MW of 25538.6 g/mol, which corresponded to the loss of 31 amino acids from the N-terminus of *Ng*PatB_Δ69_. Thus, a construct of *Ng*PatB was engineered that lacked the first 100 N-terminal amino acid residues (named *Ng*PatB_Δ100_). Similarly, the previously described construct of *Sa*OatA_Δ435_, consisting of residues 435–603, was found to experience partial degradation over extended periods of time. To improve short-term and long-term stability, a construct of *Sa*OatA_C_ consisting of residues 445–601 (*Sa*OatA_Δ445_) was engineered with an N-terminal SUMO-tag. 

Both *Sa*OatA_Δ445_ and *Ng*PatB_Δ100_ were overproduced in large quantities and purified by a combination of affinity and cation-exchange chromatographies to apparent homogeneity, as judged by SDS PAGE analysis (Appendix A, Appendix A). Yields of *Sa*OatA_Δ445_ and *Ng*PatB_Δ100_ averaged 4 and 12.5 mg per litre culture, respectively. These variants of the two PG *O-*acetyltransferases were found to be stable for over a month of storage at −20 °C (Appendix A).

### 2.2. Kinetic Parameters of NgPatB_Δ100_ and SaOatA_Δ445_

We determined the kinetic parameters of *K*_M_ and *k*_cat_ for both the *Ng*PatB_Δ100_- and *Sa*OatA_Δ445_-catalyzed hydrolysis of the fluorometric substrate 4-methylumbelliferyl-acetate (4MU-Ac). These values, together with the overall efficiency of the enzyme, as reflected by *k*_cat_/*K*_M_ values, were very similar to those reported previously for the earlier versions of the recombinant enzymes (Appendix A). Likewise, the pH-activity profile of *Ng*PatB_Δ100_, with its bell-shaped curve and pH optimum of 7.0 (Appendix A), resembled that of *Ng*PatB_Δ69_. Given this, we were able to use the two *Ng*PatB constructs interchangeably for routine assays, while using the more stable *Ng*PatB_Δ100_ for longer term experiments, such as the large-scale HTS. The pH-activity optimum of *Sa*OatA_Δ445_ was found to be 8.0 with 4MU-Ac as substrate. This differs from the pH optimum of 7.0 reported previously [41]. The reason for this difference remains unknown, but the HTS assay developed (described below) was conducted at pH 7.0 to minimize spontaneous hydrolysis of the ester-linked substrate, which occurs more rapidly in alkaline pH. 

### 2.3. Pilot High Throughput Screen for Inhibitors of SaOatA_C__Δ435_ and NgPatB_Δ69_ Esterase Activity

The assay used to monitor the transferase activity of PG *O*-acetyltransferases [41,44] involves costly glycans, as well as a laborious analysis not conducive to a high throughput format, and so it was not considered further. Instead, an in vitro microtiter plate assay to monitor the esterase activity [46] of the two enzymes was optimized for use in a high throughput format. As many of the compounds in HTS collections are yellow when dissolved in aqueous solutions, we chose to use the fluorogenic substrate 4MU-Ac, instead of the chromogenic substrate *p*-nitrophenyl acetate (*p*NP-Ac), to avoid possible interference by a colorimetric assay product. Thus, using excitation and emission wavelengths fixed at 372 nm and 445 nm for the detection of the 4MU product, we tested different combinations of enzymes and substrate concentrations in a total reaction volume of 50 µL of 50 mM sodium phosphate, pH 7.0, containing 5% DMSO. Under these conditions, the reaction of 3 µM *Ng*PatB_Δ69_ or *Sa*OatA_Δ435_ with 1 mM 4MU-Ac gave an ideal level of detectable product, following 6 min incubation at 22 °C. 

The measure of the quality and suitability of an HTS assay is typically assessed by calculating its Z’ value [47]. The Z’ is a statistical parameter which takes both the dynamic range of the signal and the data variation associated with the controls into account. A Z’ value ≥ 0.5 (to a maximum of 1.0) implies that a well-defined “hit-window” will exist and that the assay is appropriately optimized for use in an HTS. For the two pilot screens, we used 192 replicates of each the high (negative; substrate and enzyme) and low (positive; substrate only) control assays for the calculation of the final Z’. This gave values of 0.71 and 0.67 for *Ng*PatB_Δ69_ and *Sa*OatA_Δ435_, respectively, indicating that the optimized assay conditions were indeed suitable for use in an HTS (Figure 2a,c). 

To test the robustness of the assay, we conducted pilot screens of 3921 small molecules that comprise the Bioactive Subset of the Canadian Compound Collection [48] (see Appendix A for a flowchart of the process). The ability of these molecules to inhibit *Ng*PatB_Δ69_ and *Sa*OatA_Δ435_ esterase activity was tested in duplicate. The threshold to identify “hits” was set at ≤77% and ≤70% residual activity of *Ng*PatB_Δ69_ and *Sa*OatA_Δ445_, respectively, where 77% and 70% represented 3 standard deviations below the respective mean of the high control reactions on each corresponding plate (Figure 2b,d). Using this threshold, 58 and 25 unique primary hits were identified for the two respective enzymes, with the residual activity of both replicates being below the threshold. This resulted in a relatively high primary hit rate of 1.5% and 0.64% for the screens of *Ng*PatB_Δ69_ and *Sa*OatA_Δ435_, respectively.

### 2.4. Secondary Screening of Pilot Screen Hits

We selected a number of the primary hits from the two pilot screens for testing in dose response assessments to determine half maximal inhibitory concentrations (IC_50_) using 4MU-Ac as substrate (Figure 3 and Appendix A). At this point, we discarded several hits from the respective primary screens due to a lack of response to increasing concentration of compound. Several molecules that did confirm to be dose dependent, and inhibitory to both enzymes, shared a common coumarin core (Figure 3) and so we investigated them further. Of these, esculin, esculetin, and scopoletin were identified as effective inhibitors of both *Ng*PatB_Δ69_ and *Sa*OatA_Δ435_. The IC_50_ values determined for these compounds ranged from 48 µM to 106 µM (Table 1, Appendix A). In contrast, the IC_50_ values for the simpler 7-hydroxycoumarin were an order of magnitude greater. 

With fluorometric assays, fluorescence quenching by compounds in the libraries could lead to their false identification as inhibitors. Thus, to confirm their inhibitory activity, we also tested esculin, esculetin, and scopoletin as inhibitors of PatB_Δ69_ and *Sa*OatA_Δ435_ esterase activity using the chromogenic *p*NP-Ac as substrate. The resulting dose response curves confirmed that each of the compounds inhibits both enzymes. However, the apparent IC_50_ values for each of the tested coumarins were over an order of magnitude higher compared to the respective values determined using 4MU-Ac as substrate (Figure 3 and Appendix A, Table 1).

### 2.5. Large Scale High Throughput Screen for Inhibitors of NgPatB_Δ100_ and SaOatA_Δ445_

The fluorescence-based assay using 4MU-Ac as substrate was further optimized for use with the tipless Dual VaLet™ (Vertical Array Loader) HTS system of the Drug Discovery Facility at the SPARC BioCentre. Z’ scores were determined for assays with varying incubation times, as well as concentrations of each enzyme and substrate (*n* = 96 for *Ng*PatB; *n* = 64 for *Sa*OatA). For *Ng*PatB_Δ100_, a final Z’ of 0.82 was achieved with 0.75 μM enzyme incubated with 0.48 mM 4MU-Ac in a total volume of 26 μL for 12 min at 22 °C (Appendix A). Optimal conditions for *Sa*OatA_Δ445_ involved incubation of 0.75 μM enzyme with 0.2 mM 4MU-Ac in a total volume of 47 µL for 42 min at 22 °C, which gave a Z’ score of 0.85 (Appendix A). 

Using these assays, we tested a total of 134,993 compounds in singlicate for inhibition of *Ng*PatB_Δ100_ and/or *Sa*OatA_Δ445_ (Appendix A), where hits were defined as causing ≥20% inhibition combined with a z-score ≤−3 compared to respective in-plate controls (Appendix A). We tested *Ng*PatB_Δ100_ in a primary screen against a total of 68,928 compounds and we identified 423 hits, where 171 of the compounds originated from the Maybridge HitFinder Collection (hit rate of 1.2%), 248 from the ChemBridge DIVERset Collection (hit rate of 0.5%), and the remaining 4 compounds were from the subset of the GlycoNET ChemBridge Collection (hit rate of 0.1%). Rather than repeating the screening of this collection of compounds with *Sa*OatA_Δ445_, we tested it against the GlycoNET Collection of 66,065 compounds. Applying the initial hit parameters, we obtained 4944 hits with this collection, giving the very high hit rate of 7.5%. By re-evaluating both data sets using a z-score of ≤−4.05 and percent inhibition ≥ 20%, the numbers of initial hits were reduced to 329 (0.48%) and 631 compounds (0.93%) for *Ng*PatB_Δ100_ and *Sa*OatA_Δ445_, respectively.

For hit validation, we selected 960 compounds from the combined list of initial hits from the two primary screens and we tested each for inhibition against both *Ng*PatB_Δ100_ and *Sa*OatA_Δ445_ in separate screens, which were conducted in triplicate using 4MU-Ac as substrate. This reduced the number of authentic hits to 18 and 79 compounds as apparent inhibitors of *Ng*PatB_Δ100_ and *Sa*OatA_Δ445_, respectively. Finally, we examined each of these remaining compounds for reactive functional groups. Six and 58 compounds, respectively, were identified as “pan assay interference compounds” (PAINS) [50] and therefore we eliminated them from further consideration. The remaining 12 and 21 compounds were subjected to secondary screening.

### 2.6. Secondary Screening of Potential PatB Inhibitors from Large Scale HTS

To further evaluate the inhibitory potential of the 12 and 21 remaining compounds (Appendix A), we determined their IC_50_ values toward *Ng*PatB_Δ100_ and *Sa*OatA_Δ445_, respectively, using an 8-point dose response assay with 4MU-Ac as substrate. Four and 16 compounds that did not completely inhibit *Ng*PatB_Δ100_ and *Sa*OatA_Δ445_, respectively, or had non-ideal response curves, were eliminated (Appendix A). The remaining 13 hits between the two enzymes had calculated IC_50_ values of between 22 and 88 µM (Appendix A), two of which were common to both *Ng*PatB_Δ100_ and *Sa*OatA_Δ445_. These 2 inhibitors as well as 3 which were specific for *Sa*OatA_Δ445_, including compound 18420 with an IC_50_ of 86.6 µM, and two of the *Ng*PatB_Δ100_–specific inhibitors, including compound 89224 with an IC_50_ value of 22.1 µM (Figure 4), were selected for continued investigation based on their attractive physicochemical properties. However, prior to any further assessment of inhibitory activities, we determined the fluorescence quenching properties of the compounds, recognizing their potential to interfere with the fluorometric assay. This was accomplished using a fluorescence quenching assay where a fixed concentration of 4MU was titrated with increasing concentrations of the respective compounds. Compounds that reduced 4MU fluorescence in a concentration dependent manner were deemed to be quenchers. Unfortunately, five of the remaining seven compounds were determined to have interfered with the screening assay, leaving just compounds 89224 and 18420 for further analysis (Appendix A).

To confirm that each of the two remaining compounds were indeed inhibiting esterase activity and not quenching fluorescence in the fluorometric HTS assay, we determined their IC_50_ values using the colorimetric *p*NP-Ac based assay. Under the conditions employed, both were indeed found to be inhibitory, but with IC_50_ values in the mM range (Figure 4, Table 1).

### 2.7. Inhibition Kinetics of Esculetin and Compound 89224 with NgPatB

Given the high IC_50_ value obtained for *Sa*OatA_Δ445_ with compound 18420, we focused on determining the Michaelis–Menten parameters for only the inhibition of *Ng*PatB_Δ100_ esterase activity by compound 89224, as well as esculetin as a representative of a coumarin-based inhibitor. This was performed by varying concentrations of 4MU-Ac (0.04−2 mM) as acetyl substrate in the enzyme assay, while simultaneously increasing inhibitor concentration (0−125 µM for compound 89224 and 0−600 µM for esculetin). Nonlinear regression analysis of the inhibition data by compound 89224 provided a *K*_i_ value of 126 ± 19.5 µM, while the same analysis for esculetin provided a *K*_i_ value of 86.7 ± 4.7 µM. A Lineweaver–Burke plot of the 89224 inhibition data showed that they converge on the negative *x*-axis indicating that the compound is a mixed/non-competitive inhibitor of *Ng*PatB_Δ100_ (Figure 4). Moreover, the *K*_i_ and *K*_i_’ values are equal, indicating no preference of the inhibitor for the free enzyme or enzyme-substrate complex. Esculetin, on the other hand, appears to function as a competitive inhibitor (Appendix A).

### 2.8. Antibacterial Activity of Inhibitors

Using a microtiter plate assay, we tested each of the four identified inhibitors, the three coumarins from the pilot screen and compound 89224, for antibacterial activity toward the growth of *S. aureus* ATCC 6538 and/or *N. gonorrhoeae* FA1090. For this preliminary experiment, we used *E. coli* as a negative control because it both lacks a PG*-O*-acetyltransferase homolog and it does not produce O-acetylated PG [15]. Under the conditions employed, neither esculin nor scopoletin had any significant effect on the growth of *N. gonorrhoeae* FA1090 or *S. aureus* ATCC 6538 at concentrations up to their solubility limits of 2048 µg/mL and 480 µg/mL, respectively. In contrast, at a concentration of 512 µg/mL, esculetin retarded growth of *N. gonorrhoeae* FA1090 and *S. aureus* ATCC 6538 such that only 25% and 10% of control growth levels where reached over the course of the experiment, respectively (Appendix A). Due to its solubility limit, we could not achieve 100% inhibition with this coumarin, which therefore precluded our ability to determine its minimum inhibitory concentration (MIC) value in either organism. However, we found esculetin to also inhibit the growth of *E. coli* BL21, with an MIC of 512 µg/mL. Inhibition of all 3 bacteria indicates that, perhaps in addition to PatB and OatA, esculetin inhibits an essential target(s) common to these bacteria. 

Compound 89224 also inhibited the growth of *N. gonorrhoeae* FA1090. Similarly to esculetin, this compound reduced the growth achieved by cultures compared to controls lacking added compound, as measured by optical density at 600 nm. At 177 µg/mL and close to its solubility limit, compound 89224 caused a 90% reduction in growth over the time course of the experiment. In contrast to esculetin, however, we found compound 89224 did not affect the growth of *E. coli* BL21 to any extent within its solubility limits (Figure 5). 

## 3. Discussion

The *O*-acetylation of PG was first identified as an important virulence factor over a decade ago [18,19], at a time when the mechanics of the enzymatic system had yet to be investigated. In the years since, a number of studies have outlined novel protocols and characterizations of both PatB and OatA. In this study, we used these tools to design the first high throughput screens for the identification of inhibitors of *Ng*PatB and *Sa*OatA. One of these inhibitors was then used to provide preliminary proof of principal that PG *O-*acetyltransferase represents a promising antibacterial target.

Coumarins aside, the structural features of the inhibitors of *Ng*PatB and *Sa*OatA identified in both the pilot and large-scale screens varied considerably, with no obvious chemical similarities with which to categorize them. Most compounds, however, contained at least one, if not multiple, planer 6-membered ring, which likely mimics the aminosugars of the natural PG substrate of these *O*-acetyltransferases. The structural diversity of the inhibitors may also reflect the various substrate binding subsites that must be present to specifically accommodate both the carbohydrates of the glycan backbone and amino acids of stem peptides of PG. Despite the overall structural similarity between PatB, OatA, and *O-*acetyl-PG esterase (Ape), in addition to their identical mechanisms of catalytic action (recently reviewed in [12,13]), the effective competitive inhibitor of Ape, purpurin (1,2,4-trihydroxy-anthraquinone) [51], was not identified in the pilot HTS as an inhibitor of *Ng*PatB or *Sa*OatA. This anthraquinone is amongst a number of others that comprise the biologically-active collection of compounds tested against in the pilot screens. Not identifying purpurin within the hit list of OatA and PatB inhibitors is consistent with an earlier preliminary characterization of *Sa*OatA inhibition. The IC_50_ value of purpurin for *Sa*OatA is approximately 1.1 mM [52]. Thus, regardless of their overall similarities, the binding specificities of the −1 or +1 binding subsites at the catalytic centers of the two respective enzyme classes must be a little different.

Unlike most other Serine hydrolases, and PG-active enzymes such as lysozyme [53] and *N. meningitidis* Ape [54], the PG *O*-acetyltransferases lack deep clefts or pockets. Instead, the active site appears to be open and exposed on the enzymes’ surfaces [41,55]. Presumably, with the juxtaposition of the cytoplasmic membrane below, and PG strands of the sacculus above, the physical constraints imposed would serve to guide appropriate interactions with PG [41]. Unfortunately, a structure of neither OatA nor PatB in complex with any ligand has been determined and so details of substrate binding remain unknown. However, both PatB and OatA have been shown previously to have increasing transferase efficiency when presented with glycan acceptors with increasing degrees of polymerization [41,44]. This suggests that multiple binding subsites must exist, as has been seen for numerous other carbohydrate active enzymes, including lysozyme [53]. However, the in vitro assays adopted for the HTS and secondary screening employed structurally simple esters of acetate, which are very poor mimics of the natural PG substrate. These small substrates would occupy no more than the equivalent of a single PG-binding subsite and the simplistic in vitro assay conditions used would not exert any physical constraints naturally involved in the productive binding of PG. This lack of constraint, together with some potential binding flexibility, may explain the relatively large differences in IC_50_ values obtained for each inhibitor when using the two different assays (Table 1). Thus, it would appear that the relatively small and conformationally flat pseudo substrates may bind productively in subtly different ways such that the coumarin-based inhibitors more efficiently block the enzymes’ activity toward 4MU-Ac, compared to inhibiting their activity toward *p*NP-Ac. The characterization of esculetin as an apparent competitive inhibitor of *Ng*PatB, when assayed using 4MU-Ac, supports this view. Moreover, the identification of a number of coumarins as inhibitors would be a natural consequence of using 4MU-Ac as the substrate for the primary HTS assay. 

A structure-activity relationship (SAR) analysis based on comparisons of determined IC_50_ values revealed some specificity of the different coumarins as inhibitors of the PG *O-*acetyltransferases. For example, the presence of either a hydroxyl or a glucopyranosyl group at C6 and the absence of a 4-methyl group enhanced the inhibitory potential of the 7-hydroxy-coumarin based derivative toward *Ng*PatB compared to hymerchrome (7-hydroxy-4-methyl-coumarin) by over an order of magnitude (Table 1). The observation that hymercrome is a relatively poor inhibitor, however, is somewhat surprising given that it is a very close homolog of the substrate 4MU-Ac (7-acetyl-4-methy-coumarin). The reason for this apparent anomaly is not known, but intuitively it would have been expected to serve as a highly effective competitive inhibitor like esculetin. 

Coumarin (1,2-benzopyrone) is a natural product derived from the phenylpropanoid pathway. Hundreds of coumarin derivatives have been isolated from plants and fungi with many being exploited for their beneficial properties towards human health. As a versatile synthetic scaffold possessing a wide spectrum of biological effects, coumarin-based drugs are being developed to treat a variety of diseases including cancer, while warfarin (a 4-hydroxy-coumarin derivative, also known as coumadin) is used widely as an anticoagulant (recently reviewed in [56,57]). In addition, coumarinic and isocoumarinic derivatives have long been known to serve as potent inhibitors of Serine hydrolases. The mode of inhibition by these compounds, however, is irreversible, involving the initial acylation of the catalytic Ser through its nucleophilic attack on the carbonyl C atom of the respective bicyclic ring compounds [58,59,60]. With the PG *O*-acetyltransferases investigated in the current study, the inhibition was found to be independent of time and hence reversible. That the coumarins do not serve as mechanism-based inhibitors of the PG *O-*acetyltransferases is likely consistent with the observation that, at least, esculetin functions as a competitive inhibitor. Binding in the same mode as 4MU-Ac would not position the carbonyl C atom of the 1,2-benzopyrone for appropriate attack by the catalytic Ser residues. With 4MU-Ac, binding of 4MU places the sidechain acetyl carbonyl in this position.

The coumarin-based inhibitors of both PatB and OatA were found to have antibacterial activity, which is in keeping with earlier reports ([61] and references therein). However, we found that this antibacterial activity was not only confined to bacteria that produce *O-*acetyl-PG, suggesting that they have “off-target” effects. As such, we could not use them to validate the enzymes as antibacterial targets. Given the wide distribution of coumaric compounds in nature, and the range of their uses in a variety of medicinal and therapeutic practices, perhaps it is not surprising that the relatively simple coumarins identified in the current application would have off target effects, as was with the inhibition of *E. coli* growth. Compound 89224, on the other hand, has both comparable inhibitory activity toward *Ng*PatB and apparent antibacterial specificity toward *N. gonorrhoeae.* Compound 89224, a benzothiazolyl-pyrazolo-pyridine derivative, functions as a non-competitive inhibitor of *Ng*PatB and, in so-doing, inhibits the growth *N. gonorrhoeae*. *E. coli,* a bacterium that produces neither *O*-acetyl-PG nor a PatB homolog, is not affected by this compound. This distinction provides preliminary evidence that, at least, PatB represents a new antibacterial target worthy of further investigation.

Sensitizing pathogenic bacteria to the lysozyme of the innate immune system would serve to control infections. With pathogens such as *N. gonorrhoeae*, it also has the potential to alleviate the wide range of downstream complications that often result in patients following gonorrheal infections. As lysozyme hydrolyzes only the non-O-acetylated portions of the PG sacculus, large molecular weight (O-acetylated) fragments continue to circulate in the patient, a condition that leads to the development of e.g., pelvic inflammatory disease [62] and rheumatoid arthritis [28]. Similarly, in a mouse-model of *S. aureus* arthritic infection, the severity and frequency of arthritis was increased with the presence of OatA, resulting in increased articular tissue damage as a result of the decreased ability of lysozyme in the synovial tissue to kill *S. aureus* with O-acetylated PG [63]. Furthermore, during *S. aureus* infection, O-acetylation of PG limits the T-helper cell priming required to develop an effective protective response to systemic infection [64]. Thus, in addition to increasing the severity of disease, PG O-acetylation in *S. aureus* also causes downstream complications and limits the development of robust protective immunity required to prevent reinfection. Inhibitors targeting PG *O*-acetyltransferases also have the potential to work synergistically with β-lactam antibiotics. For example, increased levels of PG O-acetylation in *Streptococcus pneumoniae* cause increases in β-lactam resistance. By disrupting this pathogen’s *O*-acetyltransferase gene, cells not only become hypersensitive to lysozyme, but also to β-lactams, which had been shown to be ineffective previously [30]. The results of this study demonstrate validity of this approach and provide the basis for further inhibitor discovery. 

## 4. Materials and Methods 

### 4.1. Engineering of NgPatB

A previously described recombinant construct of *Ng*PatB from *N. gonorrhoeae* FA1090, lacking its N-terminal 69 residues (previously referred to as *Ng*PatB_Δ77_, but herein referred to as *Ng*PatB_Δ69_), was engineered to include an N-terminal His_6_-SUMO tag [39]. The gene encoding the SUMO tag was sub-cloned from the Champion™ pET-SUMO expression vector (ThermoFisher Scientific, Waltham, MA) by PCR amplification using the primers 5′-GTGGCAGTGCTAGCATGTCGGACTCAG-3′ (forward) and 5′-GCTCGAGCTCACCACCAATCTGTTCTCTGT G-3′ (reverse). The PCR amplicon was digested using appropriate restriction enzymes and ligated into similarly digested pBAD-His A plasmid containing the gene for PatB_Δ69_ using T4 DNA ligase (New England Biolabs, Ltd, Whitby, ON Canada), resulting in the pACAB1 construct. This construct was further modified by removing the restriction site between *sumo* and *NgpatB_Δ69_* to encode a flush fusion protein. Outward facing primers modified with 5′phosphate groups (5′-CTGTCCGGCGAAACTCCGC-3′, forward and 5′-ACCACCAATCTGTTCTCTGTGAGCC-3′, reverse) were used to amplify the plasmid excluding the restriction site. The resulting PCR product was ligated using T4 DNA ligase to generate the final pACAB2 construct. This construct (*Ng*PatB_Δ69SUMO_) was used in the production of *Ng*PatB_Δ69_ containing an N-terminal His_6_-SUMO fusion tag.

A second construct was generated by deleting a further 31 codons from the 5′ end of *ngpatB_Δ69_* within pACAB2. Once again outward facing primers modified with 5′phosphate groups (5′-GGCACCGAATGGAAACAGGGC-3′, forward, and 5′- ACCACCAATCTGTTCTCTGTGAGCC-3′, reverse) were used to PCR amplify the target DNA. The PCR amplicon was ligated using T4 DNA ligase resulting in the final pACAB7 construct. This construct (*Ng*PatB_Δ100SUMO_) was used in the production of a stable derivative of PatB, lacking its N-terminal 100 amino acids (*Ng*PatB_Δ100_) and possessing an N-terminal His_6_-SUMO fusion tag.

### 4.2. Production and Purification of NgPatB Variants

Competent cells of *E. coli* Rosetta (DE3) (Novagen, Millipore (Canada) Ltd., Etobicoke, ON Canada) were transformed with either pACAB2 or pACAB7 to achieve overproduction of the two engineered forms of recombinant *Ng*PatB. Cells were grown in 1 L of Super Broth (5 g sodium chloride, 20 g yeast extract, and 32 g tryptone) supplemented with 100 µg/mL ampicillin at 37 °C, with aeration until an OD_600_ of 0.6 was reached. Over expression of the genes was induced with the addition of 0.02% L-arabinose (final concentration) followed by further incubation for 4 hours before cells were harvested by centrifugation (5000× *g*, 20 min, 4 °C).

Cell pellets were resuspended in 20 mM sodium phosphate buffer, pH 7.8, containing 500 mM NaCl and 10 mM imidazole (wash buffer) and then subjected to lysis by either using a French Pressure Cell at 1200 PSI (*Ng*PatB_Δ69SUMO_) or sonication (*Ng*PatB_Δ100SUMO_). Insoluble cellular debris were cleared by centrifugation (25,000× *g*, 20 min, 4 °C) and supernatants were filtered through a 0.45 µm syringe-driven filter (Millipore). Recombinant *Ng*PatB was isolated by affinity chromatography using cOmplete His-Tag purification resin (Roche Diagnostics, Laval, QC Canada) equilibrated with wash buffer. After sample loading, the resin was washed with 10 bed volumes of wash buffer and then bound proteins were eluted in wash buffer containing 250 mM imidazole. Recovered proteins were dialyzed at ambient temperature, first against 20 mM sodium phosphate buffer, pH 7.8, containing 50 mM L-arginine, and then the same buffer lacking L-arginine.

Cleavage of the SUMO tag from recombinant *Ng*PatB was achieved by digestion with purified SUMO protease (Ulp1). SUMO protease was purified following the protocol of Reverter and Lima [65] and the fusion proteins were digested at 30 °C for 3 hours. Further purification of *Ng*PatB was achieved by cation-exchange chromatography on Source mono S (GE Healthcare Life Sciences, Mississauga, ON Canada). Samples were applied to the 10 × 100 mm column pre-equilibrated in 20 mM sodium phosphate, pH 7.8, and bound enzyme was recovered by application of a linear gradient of 0.1–0.3 M NaCl over 30 min. Purity of fractions was assessed by SDS PAGE and appropriate fractions were pooled, dialyzed against 20 mM sodium phosphate buffer, pH 7.8 at ambient temperature, and stored at −80.0 °C until required.

### 4.3. Engineering of SaOatA_Δ445_

A previously described recombinant construct of *Sa*OatA from *S. aureus* SA113, consisting of residues 435–603, was engineered to include an N-terminal His_6_- tag [41]. This gene was ligated into the Champion™ pET-SUMO expression vector using T4 DNA ligase, followed by deletion mutagenesis generating the final pDSAC71 construct. This construct consisting of residues 445–601 (*Sa*OatA _Δ445SUMO_) was used in the production of a stable derivative of OatA lacking its N-terminal 445 amino acids and possessing an N-terminal His_6_-SUMO fusion tag.

### 4.4. Production and Purification of SaOatA_Δ445_

Production and purification of *Sa*OatA_Δ445_ was carried out as follows. Competent cells of *E. coli* BL21 (DE3) (Novagen) were transformed with pDSAC71 to achieve overproduction of *Sa*OatA_Δ445SUMO_. Cells were grown in 1 L Luria-Bertani broth (Difco; Fisher Scientific, Ottawa, ON Canada) supplemented with 50 µg/mL kanamycin at 37 °C with aeration until an OD_600-_ of 0.6 was reached. Gene expression was induced with the addition 1mM isopropyl β-d-1-thiogalactopyranoside (Sigma-Aldrich Canada, Oakville, ON Canada. The culture continued to grow at 37 °C for an additional 4 h, after which the cells were harvested by centrifugation (5000× *g*, 15 min, 4 °C). 

Cell pellets were thawed on ice and resuspended in wash buffer (50 mM sodium phosphate, pH 7.5, 200 mM NaCl) containing 20 µg/mL DNase, 20 µg/mL RNase, 50 µg/mL hen egg-white lysozyme, and cOmplete EDTA-free protease inhibitor cocktail (Millipore). The cells were disrupted by sonication on ice (15 min, pulse 10:20 sec, 50% amplitude) and unbroken cells were collected by centrifugation (25,000× *g*, 20 min, 4 °C). The soluble lysate was then incubated with cOmplete His-Tag purification resin (Roche Diagnostics,) pre-equilibrated with wash buffer. After 1 h at 4 °C with nutation, the cell lysate and resin slurry were loaded onto a gravity-flow column and the cell lysate containing all unbound proteins was allowed to pass through. The resin was washed with 40 resin bed volumes of wash buffer before elution of *Sa*OatA_Δ445SUMO_ from the resin with elution buffer (50 mM sodium phosphate, pH 7.5, 200 mM NaCl, 400 mM imidazole). To achieve cleavage of the His_6_-SUMO tag from recombinant *Sa*OatA_Δ445SUMO_, purified SUMO protease (Ulp1) was added to the elution fraction. This protein mixture was dialyzed against 50 mM sodium phosphate pH 6.5, 150 mM NaCl at ambient temperature for 2 h, with one buffer change. The dialyzed fraction was then incubated for 1 h with nutation at 4 °C with fresh cOmplete His-Tag purification resin pre-equilibrated with dialysis buffer to bind the His_6_-SUMO tag. The solution containing untagged *Sa*OatA_Δ445_ was allowed to pass through the column and was collected. The *Sa*OatA_Δ445_ fraction was concentrated using an Amicon Ultra-15 centrifugal filter (10 kDa MWCO; Millipore) at 4000× *g* at 4 °C. After concentration, the protein was filtered using a syringe-driven filter (0.22 µm; Millipore) and loaded onto a HiLoad Superdex 75PG column (GE Healthcare) pre-equilibrated with gel filtration buffer (50 mM sodium phosphate pH 6.5, 150 mM NaCl). Protein elution was achieved using an isocratic flow of gel filtration buffer at a flow-rate of 1 mL/min. Purity of fractions was assessed by SDS-PAGE and appropriate fractions were pooled and stored at 4 °C until use. 

### 4.5. Assays for *Ng*PatB and *Sa*OatA Activity and Inhibition Kinetics

The ability of *Ng*PatB and *Sa*OatA variants to function as esterases was exploited for the routine measurements of their catalytic activity. Microtiter plate-based fluorometric or chromogenic assays, using either 4MU-Ac or *p*NP-Ac as substrate, were conducted as previously described [46]. Michaelis–Menten steady-state kinetic parameters were established by determining esterase rates of 3 µM *Ng*PatB in 50 mM sodium phosphate buffer, pH 7.0, or 5 µM *Sa*OatA_Δ445_ in 50 mM sodium phosphate buffer, pH 6.5, in the presence of 0.005–5 mM or 0.02–0.5 mM 4MU-Ac, respectively. The Michaelis–Menten kinetic parameters were determined by nonlinear regression analysis of the data using GraphPad Prism 5.0 (Informer Technologies, Inc., Roseau Valley, Dominica). The pH dependence of enzyme activity was determined by substituting the sodium phosphate buffer in the reaction described above with a 50 mM tripartite buffer consisting of one-third sodium citrate, one-third sodium phosphate, and one-third sodium borate at a range of pHs. The pH dependence of the reactions with *Ng*PatB and *Sa*OatA_Δ445_ was determined in the presence of 1.0 mM and 0.1 mM 4MU-Ac, respectively.

Determinations of *K*_i_ and mode of inhibition of *Ng*PatB_Δ100_ were conducted using 4MU-Ac as a substrate. Reactions involved incubating 3 µM *Ng*PatB_Δ100_ in 50 mM sodium phosphate buffer, pH 7.0, for 10 min with 0–125 µM compound 89224 or 0–600 µM esculetin prior to the addition of 0.04–2 mM 4MU-Ac. Initial velocities of substrate hydrolysis were obtained by fluorescence emission at 420 nm (390 nm excitation).

### 4.6. Pilot HTS for Inhibitors of NgPatB_Δ69_ and SaOatA_Δ435_ and Esterase Activity 

In the Pilot HTS screen 3 µM *Ng*PatB_Δ69_ or *Sa*OatA_Δ445_ (final concentration) was assayed with 1 mM 4MU-Ac as substrate in 50 mM sodium phosphate buffer (pH 7.0). The release of the fluorescent 4MU product at ambient temperature was monitored using excitation and emission wavelengths of 372 nm and 445 nm, respectively. Assessment of both the quality of the assay and readiness for use in a high throughput format was performed using a Z’ assay, the result of which was calculated using Equation (1) [47], as follows:(1)Z′ Factor = 1−[(3σh + 3σl)/(|μh − μl|)],
where σh, σl, µh, and µl are the standard deviations (σ) and averages (µ) of the high (h, 100% activity, negative) and the low (l, 0% activity, positive) controls.

A collection of 3921 small molecules (10 µM final concentration) comprising the Bioactive Subset of the Canadian Compound Collection [48] was tested against *Ng*PatB_Δ69_ and *Sa*OatA_Δ435_ activity in duplicate using a 384-well microtiter plate format (Costar 3573). A Biomek^®^ FX^P^ Laboratory Automation Workstation (Beckman Coulter Canada, Mississauga, ON, Canada) liquid handling system maintained at the HTS Facility at the Centre for Microbial Chemical Biology, McMaster University, was used to prepare all 384-well assay plates. SAMI EX (Beckman Coulter, City Country) method scheduling software permitted a 10 min pre-incubation of enzymes with the added compound prior to the addition of substrate and subsequent data collection using an EnVision^®^ 2014 Multilabel Plate Reader (Perkin Elmer, Waltham, UK). All steps were performed at ambient temperature with both enzymes having been stored at 4 °C prior to its addition to assay plates. “Hits” were identified as any compound for which both replicates had a lower residual activity than the identified threshold, which was set as 3 standard deviations below the mean of the high (100% activity) control.

### 4.7. Secondary Screening of Potential NgPatB Inhibitors from Pilot HTS 

Secondary screening involved the determination of IC_50_ values for a select number of hits identified in the pilot screen. PatB_Δ69_ or *Sa*OatA_Δ445_ (3 µM) was assayed with 1 mM 4MU-Ac in 50 mM sodium phosphate buffer, pH 7.0, in the presence of 0.001–2.0 mM concentrations of inhibitors. Data were fit to the [Inhibitor] vs. Normalized response – Variable slope (Equation (2)) within GraphPad Prism 5.0.
(2)$$Y\;=\;100/\left(1+\frac{x^{\mathrm{HillSlope}}}{\mathrm{IC}50^{\mathrm{HillSLope}}}\right).$$

### 4.8. Large Scale High Throughput Screen for Inhibitors of *Ng*PatB_Δ100_ and *Sa*OatA_Δ445_ Esterase Activity

A large scale HTS was conducted at the SPARC BioCentre Drug Discovery Facility (The Hospital for Sick Children, Toronto, Canada) to identify additional inhibitors of PatB_Δ100_ acetylesterase activity. A total of 68,928 compounds were tested in singlicate, including 14,400 from the Maybridge HitFinder Collection, 50,000 from the ChemBridge DIVERset Collection, and 4,480 compounds representing plates 1–15 of the GlycoNET ChemBridge Collection maintained at the facility. The 4MU-Ac served as the substrate in the 384-well microtiter plate (NUNC 262260) fluorescent assay described above. The assay was optimized for use with this facility by altering enzyme and substrate concentrations, in addition to incubation times and reaction volume. As for the pilot screen, a Z’ for the assay was determined using Equation (1). The optimized assay involved dispensing a final concentration of 0.75 µM PatB_Δ100_ in 50 mM sodium phosphate buffer, pH 7.0, into plate wells using a Thermo MultiDrop Combi fitted with a standard cassette. This was followed by dispensing final concentrations of 20 µM compounds from the Maybridge HitFinder and ChemBridge DIVERset Collections or 10 µM compounds from the GlycoNET ChemBridge Collection using an Echo^®^ 550 acoustic dispenser (Labcyte). Following 10 min pre-incubation at ambient temperature, 4MU-Ac was added using a Thermo MultiDrop Combi, fitted with a small cassette, to a final concentration of 0.48 mM. Fluorescence was monitored with a SynergyNeo microplate reader (BioTek, Winooski, VT USA) using excitation and emission wavelengths of 390 and 420 nm, respectively. 

The large scale HTS for inhibitors of *Sa*OatA_Δ445_ was conducted as described above, with the following changes: Compounds were pre-incubated with *Sa*OatA_C_ at a final concentration of 0.75 µM for 40 min, followed by addition of 0.2 mM 4MU-Ac. A total of 66,065 compounds were tested in singlicate from the GlycoNET ChemBridge Collection. 

Data analysis was performed using CDD Vault Analysis Software (Collaborative Drug Discovery, Inc., Burlingame, CA, USA) to assess the effect of each compound by comparing the % residual activity of each assay sample vs. that of the in-plate high control (0.75 µM *Ng*PatB_Δ100_ or *Sa*OatA_Δ445_ in the presence of DMSO, in the place of compound). Data were normalized to the high control on a per plate basis. Hits were identified as any compound for which both threshold criteria were met (≥20% inhibition and a z-score ≤ −3.0). The z-score is a comparison of the deviation of a signal on a well by well basis, compared to the average signal of the control on the respective plate. A z-score of e.g., −3.0 represents a datum being 3.0 standard deviations below the mean of the control data.

Hits identified using these criteria were then validated in triplicate using the in vitro assay described above. The resulting list of confirmed hits were then examined and compounds which were expected to be reactive or contain functional groups identified as “pan assay interference compounds” (PAINS) [50] were eliminated.

### 4.9. Secondary Screening of Potential NgPatB and SaOatA_Δ445_ Inhibitors from Large Scale HTS

IC_50_ values were determined for a select number of hits identified in the primary screen from dose response curves generated using 0.5–160 µM (final concentrations) of respective compounds. Residual activity of *Ng*PatB_Δ100_ (0.75 µM) was assayed using 0.46 mM 4MU-Ac in 50 mM sodium phosphate buffer, pH 7.0, following 12 min incubation at ambient temperature (within the linear range of the assay). Residual activity of *Sa*OatA_Δ445_ (0.75 µM) was assayed using 0.2 mM 4MU-Ac in 50 mM sodium phosphate buffer, pH 7.0, following 40 min incubation at ambient temperature. Percent residual activity was calculated relative to the mean of the high controls from the respective plate, and the IC_50_ inhibition plots were fit using CDD Vault Analysis Software. 

Hits which were seen to affect *Ng*PatB_Δ100_ esterase activity in a dose dependent manner were also tested as inhibitors of the enzyme (0.75 µM, final concentration) using 2 mM *p*NP-Ac as substrate to identify false positives, due to fluorescence quenching under similar assay conditions. Similarly, hits which were seen to affect *Sa*OatA_Δ445_ esterase activity in a dose dependent manner were also tested as inhibitors of the enzyme at 1.5 µM (final concentration) using 1.1 mM *p*NP-Ac. Data were fit to the inhibitor vs. normalized response – variable slope equation within GraphPad Prism 5.0, as described above.

To determine whether a compound quenched the fluorescence of 4MU, its fluorescence was measured in the presence of varying concentrations of compound. Three concentrations of compounds, increasing by a factor of 10 (when possible within solubility limits), were mixed with 50 mM sodium phosphate, pH 7, with 4MU at a final concentration of 70 µM. Fluorescence was measured using a SynergyNeo plate reader (BioTek) with excitation and emission wavelengths of 390 nm and 420 nm, respectively.

### 4.10. Bacterial Strains and Growth Conditions for Minimum Inhibitory Concentration Assays

A single colony of *N. gonorrhoeae* FA1090, cultured on solid GC agar at 37 °C in a humid 5% CO_2_ incubator, was used to inoculate 5 mL of GCBL broth in a 10 mL culture tube supplemented with sodium bicarbonate [66]. The tube was sealed and the culture was incubated at 37 °C, with slight agitation, overnight to provide the inoculum for MIC assays. *E. coli* BL21 cells were cultured overnight at 37 °C with agitation (200 rpm) in 5 ml tryptic soy broth. 

MIC assays were conducted in triplicate using sterile 96-well microtiter plates (Costar 3997). Assays with *N. gonorrhoeae* were conducted using GCBL broth supplemented with sodium bicarbonate with light agitation. The microtiter plate lid was used in these experiments to maintain an internal atmospheric environment, with increased levels of CO_2_ throughout the experiment. Experiments with *E. coli* BL21 were conducted in Luria-Bertani broth with agitation and contamination was prevented with the use of Breathe Easy Sealing Membranes (Sigma-Aldrich). 

Appropriate dilutions of overnight cultures in their respective liquid growth media were incubated with two-fold dilutions of compound and growth was monitored at 600 nm over the course of an overnight incubation at 37 °C with agitation using a Synergy H1 Microplate Reader (BioTek). The growth, measured as difference in OD_600_ reading, of each treatment was then compared to that of the no compound control and plotted as a function of compound concentration. All assays were considered complete when the stationary phase was reached by the control culture lacking compound.

## 5. Conclusions

The O-acetylation of peptidoglycan observed in pathogenic Gram-negative and Gram-positive bacteria is a key virulence factor employed to enable increased pathogenicity and resistance to lysozyme of the innate immune system [10,18,19,20,22,23,24,25,26,28]. The data presented herein represents the first identification of inhibitors of PatB and OatA. We demonstrated that inhibiting *Ng*PatB could be used as a novel antivirulence strategy, as compound 89224 was able to selectively inhibit the growth of *N. gonorrhoeae* over a non-O-acetylating control bacterium. This information may enable the development of novel therapeutics to not only treat gonococcal infections but a wide range of bacterial infections where the causative agent is an O-acetylating bacterial pathogen. Moreover, this study leads the discussion of the mode of substrate interaction for this family of transferases found to have an unconventional shallow active site with respect to other carbohydrate-active enzymes. This information can be used to gain a better understand of PG O-acetylation systems in Gram-negative and Gram-positive pathogens, as well as other O-acetylating enzymatic systems, and how they could be exploited as novel antibacterial targets.

## Figures and Tables

**Figure 1 antibiotics-08-00065-f001:**
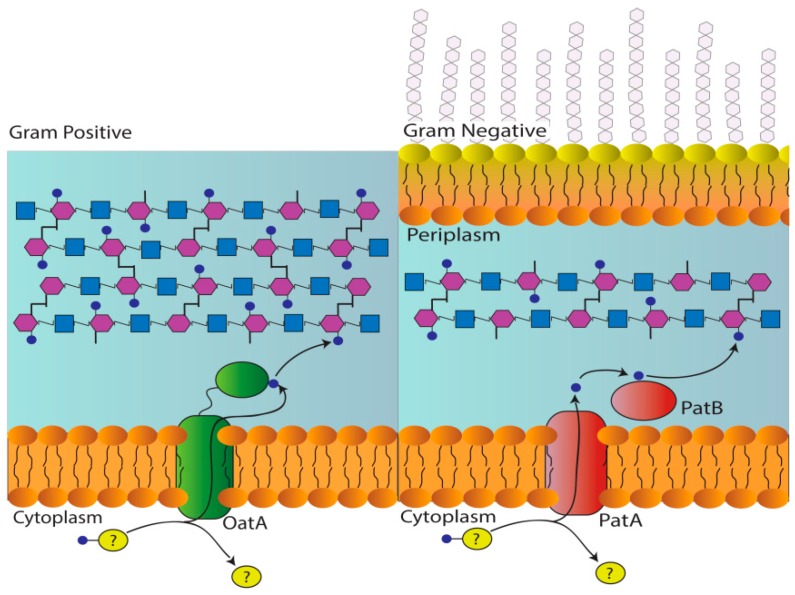
Pathways for the O-acetylation of PG. In Gram-positive bacteria, the membrane-spanning N-terminal domain of OatA receives an acetyl group from an unknown donor and translocates it across the cytoplasmic membrane for its transfer to PG by the extracytoplasmic C-terminal *O-*acetyltransferase domain. With Gram-negative bacteria, the two reactions are catalyzed by separate proteins. As an integral membrane protein, PatA translocates the acetyl group across the cytoplasmic membrane, while cytoplasmic PatB catalyzes its transfer to PG.

**Figure 2 antibiotics-08-00065-f002:**
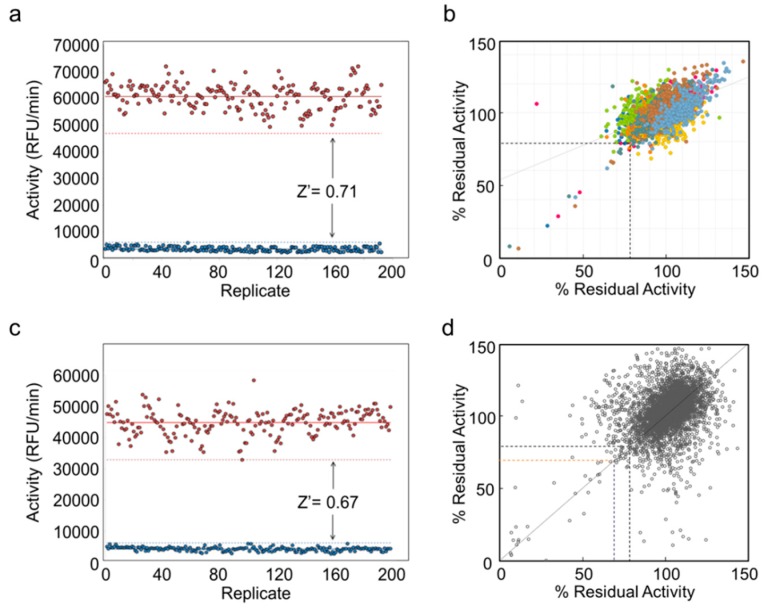
Pilot screens for inhibitors of *Ng*PatB and *Sa*OatA. (**a**,**c**) Determination of Z’ factors for the HTS assays of *Ng*PatB_Δ69_ and *Sa*OatA_Δ435_, respectively. (**b,d**) HTS of Bioactive Subset of the Canadian Compound Collection for inhibitors of *Ng*PatB_Δ69_ and *Sa*OatA_Δ435_, respectively.

**Figure 3 antibiotics-08-00065-f003:**
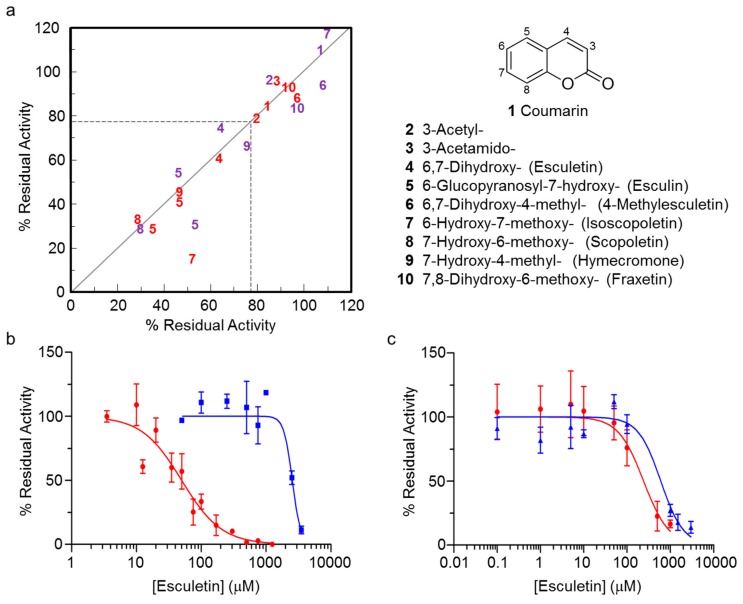
Pilot HTS for inhibitors of *Ng*PatB and *Sa*OatA. (**a**) Hit window of primary screen presenting the inhibition of *Ng*PatB_Δ69_ (red) and *Sa*OatA_Δ435_ (purple) by select coumarins present in the Bioactive Collection. Determination of dose dependence (IC_50_ values) of inhibition of (**b**) *Ng*PatB_Δ69_ and (**c**) *Sa*OatA_Δ435_ by esculetin (compound 4 in panel (**a**)), assayed with 4MU-Ac (red) and *p*NP-Ac (blue). Error bars denote standard deviation (*n* = 3).

**Figure 4 antibiotics-08-00065-f004:**
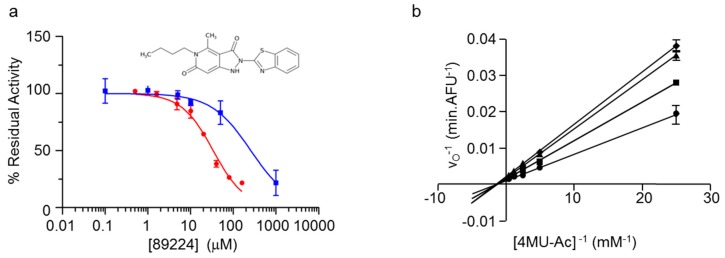
Inhibition of *Ng*PatB by compound 89224. (**a**) Determination of IC_50_ value. (**b**) Determination of the mode of reversible inhibition and *K*_i_ value. Double reciprocal plot of initial velocity **vs**. substrate concentration in the presence of (●) 0 µM, (■) 25 µM, (▲) 75 µM, and (♦) 125 µM compound 89224. Error bars denote standard deviation (*n* = 3).

**Figure 5 antibiotics-08-00065-f005:**
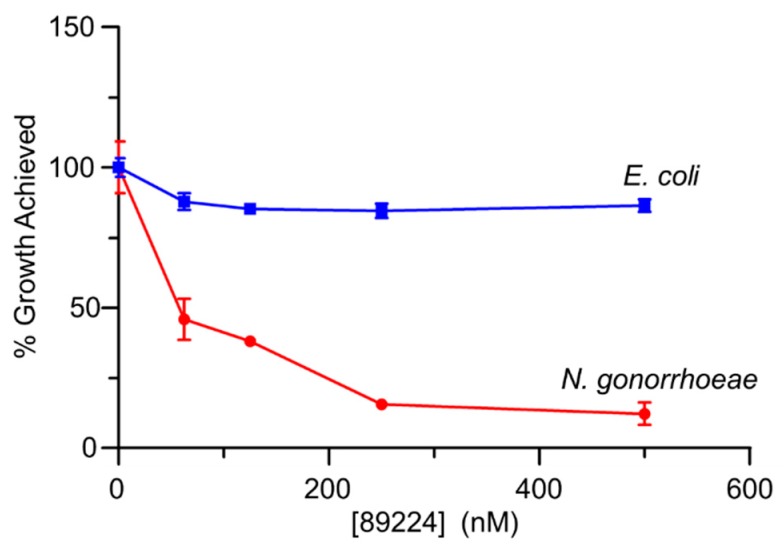
Effect of compound 89224 on growth of *N. gonorrhoeae* and *E. coli.* Cultures of the two bacteria in their respective liquid growth media were incubated at 37 °C, with compound 89224 at the concentrations indicated, and growth was monitored at 600 nm until control cultures reached stationary phase. The data presented represent the average extents of growth of cultures relative to the respective control cultures. Error bars denote standard deviations (*n* = 3).

**Table 1 antibiotics-08-00065-t001:** Inhibition of *Ng*PatB and *Sa*OatA by select compounds identified though HTS.

Compound	*Ng*PatB				*Sa*OatA		
	4MU-Ac			*p*NP-Ac	4MU-Ac		*p*NP-Ac
	1° Screen (% RA) ^1^	IC_50_ (µM)	*K*_i_ (µM)	IC_50_ (µM)	1° Screen (% RA) ^1^	IC_50_ (µM)	IC_50_ (µM)
Coumarin	82.5	n.d.	n.d.	n.d.	106	n.d.	n.d.
7-Hydroxy- (Umbelliferone)	n.d.	530 ± 70	n.d.	>3000	n.d.	98.5 ± 1.1^2^	>3000
7-Hydroxy-4-methyl- (Hymercrome)	42.9	615 ± 10	n.d.	>3000	67.8	>3000	>3000
6,7-Dihydroxy- (Esculetin)	55.7	53.1 ± 1.0	86.7 ± 4.7	853 ± 79	71.0	106±1.2^2^	600 ± 140
6,7-Dihyroxy-4-methyl- (4-Me-esculetin)	92.0	n.d.	n.d.	n.d.	98.3	n.d.	n.d.
6-Glucopyranosyl-7- hydroxy-(Esculin)	38.4	49.7 ± 3.0	n.d.	923±9.0	47.8	75.8 ± 1.0^2^	1350 ± 150
6-Hydroxy-7-methoxy- (Isoscopoletin)	53.5	n.d.	n.d.	n.d.	115	n.d.	n.d.
7-Hydroxyl-6-methoxy- (Scopoletin)	30.0	47.9 ± 4.0	n.d.	792 ± 7.0	21.4	103 ± 1.1^2^	1170 ± 100
Compound 89224	78.8	22.1 ± 1.2	126 ± 19	623 ± 64	99.8	n.d.	n.d.
Compound 18420	77.1	n.d.	n.d.	n.d.	78.2	86.6 ± 8.7	>3000

^1^*Ng*PatB data ± standard error (*n* = 3); *Sa*OatA data ± standard deviation (*n* = 3). For the coumarin data obtained from the pilot screens, average of the two replicates is presented. n.d., not determined. Data from [49].

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
