# Peer review of "Development of a High Throughput Screen for the Identification of Inhibitors of Peptidoglycan O-Acetyltransferases, New Potential Antibacterial Targets"

_antibiotics, 2019, doi:10.3390/antibiotics8020065_

Round 1

Reviewer 1 Report

Comments to author

The manuscript by Brott et al has described a high throughput screen for identification of inhibitors of peptidoglycan O- acetyltransferase (NgPatB and SaOatAc).  Overall, I find this manuscript interesting and well written. However, in my opinion, some of the conclusions are rather speculative and have no support from the data.
The manuscript needs further control experiments or not well represented,  to have a better conclusion.
I have some suggestions with respect to the presentation and control experiments, incorporation of which will help the reader to understand in a better way.

 My comments are listed below.

Major comments:

1.     The main concern is about the specificity of the inhibitors reported in the manuscripts and need further validation. As noted by the authors in line 280-287 that E. coli which lacks the enzymes was also inhibited by esculetin. The author should check the effect of the inhibitor on S. aureus oatA deletion mutant (reference 19) and see whether the deletion mutation becomes more sensitive to the inhibitors reported by the authors? This will directly show the in vivo effect of inhibitors on the O- acetyltransferase and in vitro data will be well supported. Both figure S7 and figure 5 should have a control curve for both E. coli and N. gonorrhoeae without the presence of inhibitors (just a growth media with the solvent used to dissolve the inhibitors).

2.     The conclusion from figure 5, in line number 292-293 and in the abstract 27-28 (unaffected part) seems to be misinterpreted. The author mentions “the compound 89224 did not affect the growth of E. coli BL21”. However, looking at the growth curve data it appears the inhibitors, if not inhibiting the growth, it is acting like bacteriostatic for E. coli. This can again be distinguished if the author will include a control curve for these organisms without the inhibitors.

3.     For figure 3, the author should also show a negative control ( a compound from the list which doesn’t show any inhibitory effect on PatB or OatA) and mention in the figure legend.

4.     As pointed by the author, in Line 75-76 about the synergistic effect with existing beta-lactams antibiotics. The authors should try to check the synergistic effect with inhibitors in combination with Beta-lactams. This will add up to the significance and implication of the current study.

5.     The pH-activity optimum of SaOatA∆445 was found to be 8.0 with 4MU-Ac as substrate (Line 143-145). However, a different study by Sychantha et all ( plos pathogen 2017) showed SaOatA∆435 had pH activity optimum to be 7 with p NP-Ac as substrate. The author should at least comment on this in their paper what could be the reason?

Minor comments:

1.     For section 2.5 and 2.6, the author should use a separate figure using a flowchart and arrow system to describe their High Throughput Screen for inhibitors. This will help the reader to understand the screening process a better way.

2.     The line 238-241 is confusing or phrased in not a proper way. The author mentioned that the remaining 13 hits showed the IC50 values between 22 and 88uM and referred to figure 4 and S5. But figure 4 is about compound 89224 and S5 is about the false positives. I as a reader found it highly confusing what authors want to point out and which 13 hits they are referring?

3.     For line 245-250, the author should refer to figure S5 whether they have checked the compounds that interfered with the fluorometric assay. The figure S5 legend should mention which substrate was used.

4.     There are some minor grammatical errors (like the omission of commas at many places) careful proofreading will help.

Author Response

1. The main concern is about the specificity of the inhibitors reported in the manuscripts and need further validation. As noted by the authors in line 280-287 that E. coli which lacks the enzymes was also inhibited by esculetin. The author should check the effect of the inhibitor on S. aureus oatA deletion mutant (reference 19) and see whether the deletion mutation becomes more

sensitive to the inhibitors reported by the authors? This will directly show the in vivo effect of inhibitors on the O- acetyltransferase and in vitro data will be well supported. Both figure S7 and figure 5 should have a control curve for both E. coli and N. gonorrhoeae without the presence of inhibitors (just a growth media with the solvent used to dissolve the inhibitors).

As we state on lines 290-292 (of the revised manuscript), we found that while esculetin serves as a competitive inhibitor of the esterase activity of PatB toward 4MU-Ac as substrate, its effect on bacterial growth was not restricted to species that produce O-acetylated peptidoglycan, such as E. coli. Given this “off-target” effect, the utility of this coumarin to demonstrate any importance of PatB as an antibacterial target was compromised. Consequently, we did not pursue any further study of esculetin, and, with all due respect to the reviewer, to do so now we feel would be superfluous.

2. The conclusion from figure 5, in line number 292-293 and in the abstract 27-28 (unaffected part) seems to be misinterpreted. The author mentions “the compound 89224 did not affect the growth of E. coli BL21”. However, looking at the growth curve data it appears the inhibitors, if not inhibiting the growth, it is acting like bacteriostatic for E. coli. This can again be distinguished if the author will include a control curve for these organisms without the inhibitors.

From the comments made, we believe that the reviewer has misunderstood the data presented in Figure 5. These data do not present growth curves, but rather the results of a series of growth curves generated in the presence of the compound at the concentrations indicated. So with E. coli, triplicate cultures were grown in the presence of up to 500 micromolar compound, and 100 % growth of bacteria to their normal stationary phase was observed (and reported as 100 % growth achieved). As such, no inhibition of growth (including any extended lag phase) of E. coli was observed, while N. gonorrhoeae cultures were clearly affected by the presence of this compound, leading us to conclude that these preliminary data support our proposal that this O-acetyltransferase does represent a valid antibacterial target.

3. For figure, the author should also show a negative control ( a compound from the list which doesn’t show any inhibitory effect on PatB or OatA) and mention in the figure legend.

Again, with all due respect to the reviewer, but to be frank, we do not understand the need to show that a compound that was not identified to be an inhibitor in the primary screen does not inhibit the enzymes in a dose-dependent manner. What purpose would this serve? With those compounds that do inhibit, we demonstrate the enzymes retain 100% activity until they are presented with sufficient concentrations that effect inhibition (hence a positive control for activity is intrinsic to each determination). We do not understand how showing the lack of any inhibition by a random compound provides any control for those that do inhibit. Consequently, we did not provide the requested proposed negative control.

4. As pointed by the author, in Line 75-76 about the synergistic effect with existing beta-lactams antibiotics. The authors should try to check the synergistic effect with inhibitors in combination with Beta-lactams. This will add up to the significance and implication of the current study.

As indicated above, we agree that including a study on any synergistic effects of the compound may have on the effects of different antibiotics would be interesting and of value. However, we feel that this research extends the scope of the current study which is to present our development and initial application of a new screen for inhibitors of peptidoglycan O- acetyltransferases. The submission of this manuscript was made in response to an invitation made by the editors of the Journal to contribute to a special addition entitled “Chemical Tools for Antibiotics Research”. We accepted this invitation with the prior understanding from the Editor that a manuscript presenting the development of this assay would be appropriate for this special edition. However, upon submission, the Journal requested that we move the manuscript to the current special addition entitled “Novel Strategies Against Pathogenic Bacteria”, which may have a slightly different focus. Regardless, we are not in a position at this time to expand the manuscript in the direction suggested by the reviewer. We trust that the Journal understands and appreciates our position on this issue, and perhaps may wish to return this manuscript to the former special issue.

5. The pH-activity optimum of SaOatA?445 was found to be 8.0 with 4MU-Ac as substrate (Line 143-145). However, a different study by Sychantha et all ( plos pathogen 2017) showed SaOatA?435 had pH activity optimum to be 7 with p NP-Ac as substrate. The author should at least comment on this in their paper what could be the reason?

As noted by the reviewer, we did indeed determine the pH-activity maximum for the OatA variant used in the current study to be 8, a whole unit above that previously reported for a slightly larger variant of the enzyme. At present we do not understand the reason for this difference. Regardless as suggested by the reviewer, the revised version of the manuscript provides a statement noting the difference and stating that the reason for it remains unknown

(lines 145-146).

Minor comments:

1. For section 2.5 and 2.6, the author should use a separate figure using a flowchart and arrow system to describe their High Throughput Screen for inhibitors. This will help the reader to understand the screening process a better way.

We appreciate the difficulty a reader might have in following the respective processes for the pilot and larger high throughput screens involving the two distinct enzymes and so, as suggested, we have now provided the two flowcharts in a Supplemental figure (Fig. S3), and renumbered the remaining supplemental figures accordingly.

2. The line 238-241 is confusing or phrased in not a proper way. The author mentioned that the remaining 13 hits showed the IC50 values between 22 and 88uM and referred to figure 4 and S5. But figure 4 is about compound 89224 and S5 is about the false positives. I as a reader found it highly confusing what authors want to point out and which 13 hits they are referring?

We have edited the text to hopefully make more clear the process followed and it outcomes.

3. For line 245-250, the author should refer to figure S5 whether they have checked the compounds that interfered with the fluorometric assay. The figure S5 legend should mention which substrate was used.

We have added the citation to (now) figure S6, and added appropriate text to its legend that identifies the substrate used, and meaning of the error bars.

4. There are some minor grammatical errors (like the omission of commas at many places) careful proofreading will help.

The manuscript has been checked for grammatical errors. Again, we thank the reviewer for this review.

Reviewer 2 Report

This manuscript describes the design of a high throughput method to find inhibitors for the peptidoglycan O-acetyltransferases PatA/B and OatA. The manuscript is well written and the conclusions are supported by the data. My only issue with this contribution is the low MICs obtained for the lead compounds. If peptidoglycan O-acetyltransferases are really to serve as a useful antibacterial targets, then the activity data should be more convincing. Could the authors supplement their MIC data by doing MICs with currently used antibiotics? If a synergistic effect or recovery of activity is demonstrated, that would get people excited about these enzymes.

Other points to address:

- throughout the manuscript ug/mL is written as ug≌mL

- the values in the tables and throughout the manuscript should be reported with the correct significant figures (e.g. in Table S1: 79 +/- 9, etc).

- the long term stability of the constructs should be shown by SDS-PAGE

- an explanation should be given why the construct was previously delta77 and is now delta69.

- an explanation should be given on how the error bars in Table S1 were determined.

- a sentence should be added after line 165-166 explaining how the Z' values of 0.71 and 0.67 are meaningful (e.g. an example from the literature could be given).

Author Response

This manuscript describes the design of a high throughput method to find inhibitors for the peptidoglycan O-acetyltransferases PatA/B and OatA. The manuscript is well written and the conclusions are supported by the data. My only issue with this contribution is the low MICs obtained for the lead compounds. If peptidoglycan O-acetyltransferases are really to serve as a useful antibacterial targets, then the activity data should be more convincing. Could the authors supplement their MIC data by doing MICs with currently used antibiotics? If a synergistic effect or recovery of activity is demonstrated, that would get people excited about these enzymes.

The issue of the purpose and scope of this manuscript was raised by Reviewer 1 and so rather than reiterating our response here, we refer to Major point 4 discussed above. In short, despite the validity of the issue, we believe that an extension of the research in the direction suggested by the reviewer is beyond the intended scope of the current manuscript.

Other points to address:

- throughout the manuscript ug/mL is written as ug?mL

We thank the reviewer for catching the issue of the appearance of an incorrect symbol. It would appear that an incorrect conversion was made by the different versions of Word used in saving the original manuscript as a pdf file. We have reviewed the manuscript to ensure that the correct symbol is present in the original MS doc.

- the values in the tables and throughout the manuscript should be reported with the correct significant figures (e.g. in Table S1: 79 +/- 9, etc).

Again, we thank the reviewer for catching our oversight regarding the use of significant figures. These have been corrected in the revised manuscript.

- the long term stability of the constructs should be shown by SDS-PAGE

The fact of the matter is, the samples of purified proteins presented in Figure S1 had been stored for over a month at -20 degrees C. The issue of stability is now made clear in the text of the revised version of the manuscript (lines 134-135, and in the legend to figure S1.

- an explanation should be given why the construct was previously delta77 and is now delta69.

The text now provides an explanation of the numbering change that concerned a revised annotation of the hypothetical enzyme which involved a corrected start.

- an explanation should be given on how the error bars in Table S1 were determined.

This has now been added to the table legend.

- a sentence should be added after line 165-166 explaining how the Z' values of 0.71 and 0.67 are meaningful (e.g. an example from the literature could be given).

We appreciate the concern of the reviewer, but given we provided an explanation of an acceptable Z’ factor earlier in the manuscript (current lines 162-164), we have not reiterated it here.

Reviewer 3 Report

A very fine study has produced a fine manuscript

However, let me ask you some questions.
1) Why was no screening done using a bioassay method such large plate diffusion by using a different way of solubilization?

I do not understand what the problems of solubility may have been since the IC50 tests were conducted

2) Why was there no method described for the MIC? Why have not used standard medium? Has pH any influence on the activity?

3) What are compounds 89224 and 18420, why are there no structural formulas?

Author Response

A very fine study has produced a fine manuscript

We thank the reviewer for this recognition.

However, let me ask you some questions.

1) Why was no screening done using a bioassay method such large plate diffusion by using a different way of solubilization?

This manuscript presents the development of an assay specifically for the screening of compounds as inhibitors of the peptidoglycan O-acetyltransferases OatA and PatB. The identification of any inhibitors would provide tools for the further study of the structure and function relationship of these enzymes, as wells provide an opportunity to validate them as antibacterial targets. If we understand the issue of the reviewer and the nature of a potential bioassay suggested by the reviewer, we respectfully suggest that it would not accomplish what we originally intended.

I do not understand what the problems of solubility may have been since the IC50 tests were conducted

The determination of IC50 values for enzyme inhibition required less compound in the simple in vitro assays compared to the concentrations required for bacterial cell growth studies. To compound this, the considerably higher solute concentration of the culture media compared to the simple buffer of an enzyme assay solution serves to decrease the solubility limit.

2) Why was there no method described for the MIC?

A method for the determination of the effect of the compounds on cell growth was/is described in section 4.8 Bacterial Strains and Growth Conditions for Minimum Inhibitory Concentration Assays.

Why have not used standard medium?

We are not sure what “standard medium” the reviewer is referring to, but as N. gonorrhoeae is a highly fastidious bacterium, we are limited to specific media for its culture. The medium used in this study is that commonly used for the culture of this bacterium. 

Has pH any influence on the activity?

Likewise, the pH of the medium has to be maintained at 7.4 for growth of N. gonorrhoeae and hence a study of the effect of pH on any inhibitory effect of the compound on cell growth is not possible.

3) What are compounds 89224 and 18420, why are there no structural formulas?

The structure of compound 8224 was/is presented in Figure 4a. We did not present the structure of 18420 because we established that it was not an effective inhibitor and of no use for further study (as stated on line 261 of the original manuscript).

Round 2

Reviewer 1 Report

1. The main concern is about the specificity of the inhibitors reported in the manuscripts and need further validation. As noted by the authors in line 280-287 that E. coli which lacks the enzymes was also inhibited by esculetin. The author should check the effect of the inhibitor on S. aureus oatA deletion mutant (reference 19) and see whether the deletion mutation becomes more

sensitive to the inhibitors reported by the authors? This will directly show the in vivo effect of inhibitors on the O- acetyltransferase and in vitro data will be well supported. Both figure S7 and figure 5 should have a control curve for both E. coli and N. gonorrhoeae without the presence of inhibitors (just a growth media with the solvent used to dissolve the inhibitors).

As we state on lines 290-292 (of the revised manuscript), we found that while esculetin serves as a competitive inhibitor of the esterase activity of PatB toward 4MU-Ac as substrate, its effect on bacterial growth was not restricted to species that produce O-acetylated peptidoglycan, such as E. coli. Given this “off-target” effect, the utility of this coumarin to demonstrate any importance of PatB as an antibacterial target was compromised. Consequently, we did not pursue any further study of esculetin, and, with all due respect to the reviewer, to do so now we feel would be superfluous.

Reviewer comment: Satisfied with the author comment.

2. The conclusion from figure 5, in line number 292-293 and in the abstract 27-28 (unaffected part) seems to be misinterpreted. The author mentions “the compound 89224 did not affect the growth of E. coli BL21”. However, looking at the growth curve data it appears the inhibitors, if not inhibiting the growth, it is acting like bacteriostatic for E. coli. This can again be distinguished if the author will include a control curve for these organisms without the inhibitors.

From the comments made, we believe that the reviewer has misunderstood the data presented in Figure 5. These data do not present growth curves, but rather the results of a series of growth curves generated in the presence of the compound at the concentrations indicated. So with E. coli, triplicate cultures were grown in the presence of up to 500 micromolar compounds, and 100 % growth of bacteria to their normal stationary phase was observed (and reported as 100 % growth achieved). As such, no inhibition of growth (including any extended lag phase) of E. coli was observed, while N. gonorrhoeae cultures were clearly affected by the presence of this compound, leading us to conclude that these preliminary data support our proposal that this O-acetyltransferase does represent a valid antibacterial target.

Reviewer comment: I think the author should explain it better in the figure legend by incorporating the statement made above to avoid misinterpretation. I agree with the author that N. gonorrhoeae show a sharp decrease in % growth achieved as the concentration of inhibitor is increased. However, for E. coli the second point as compared to the starting point for the % growth achieved also show a significant drop and then become constant. What could be the reason for that drop from the first point to the second point (% growth achieved)? The author should comment on this.

3. For figure, the author should also show a negative control ( a compound from the list which doesn’t show any inhibitory effect on PatB or OatA) and mention in the figure legend.

Again, with all due respect to the reviewer, but to be frank, we do not understand the need to show that a compound that was not identified to be an inhibitor in the primary screen does not inhibit the enzymes in a dose-dependent manner. What purpose would this serve? With those compounds that do inhibit, we demonstrate the enzymes retain 100% activity until they are presented with sufficient concentrations that effect inhibition (hence a positive control for activity is intrinsic to each determination). We do not understand how showing the lack of any inhibition by a random compound provides any control for those that do inhibit. Consequently, we did not provide the requested proposed negative control.

Reviewer comment: My suggestion was for the better representation of the result shown by the author and interpretation by the readers ( for the overall improvement of the manuscript). As mentioned by the author in line 189-190 “with fluorometric assays, fluorescence quenching by compounds could lead to their false identification as inhibitor”. In my opinion, any compound from the list that does not inhibit the enzyme (negative control) but has the same coumarin core will help in distinguishing the false positives from the actual inhibitor or author can just include the fluorescence curve of inhibitor compound above its inhibitory concentration without enzyme and show that inhibitor does not quench fluorescence by itself above its inhibitory concentration. The negative control as a separate panel in figure 3 will further validate the fluorometric assay used by the author and help in comparing non-inhibitors with the actual inhibitor.

4. As pointed by the author, in Line 75-76 about the synergistic effect with existing beta-lactams antibiotics. The authors should try to check the synergistic effect with inhibitors in combination with Beta-lactams. This will add up to the significance and implication of the current study.

As indicated above, we agree that including a study on any synergistic effects of the compound may have on the effects of different antibiotics would be interesting and of value. However, we feel that this research extends the scope of the current study which is to present our development and initial application of a new screen for inhibitors of peptidoglycan O- acetyltransferases. The submission of this manuscript was made in response to an invitation made by the editors of the Journal to contribute to a special addition entitled “Chemical Tools for Antibiotics Research”. We accepted this invitation with the prior understanding from the Editor that a manuscript presenting the development of this assay would be appropriate for this special edition. However, upon submission, the Journal requested that we move the manuscript to the current special addition entitled “Novel Strategies Against Pathogenic Bacteria”, which may have a slightly different focus. Regardless, we are not in a position at this time to expand the manuscript in the direction suggested by the reviewer. We trust that the Journal understands and appreciates our position on this issue, and perhaps may wish to return this manuscript to the former special issue.

Reviewer comment: Satisfied with the author comment.

5. The pH-activity optimum of SaOatA?445 was found to be 8.0 with 4MU-Ac as substrate (Line 143-145). However, a different study by Sychantha et all ( plos pathogen 2017) showed SaOatA?435 had pH activity optimum to be 7 with p NP-Ac as substrate. The author should at least comment on this in their paper what could be the reason?

As noted by the reviewer, we did indeed determine the pH-activity maximum for the OatA variant used in the current study to be 8, a whole unit above that previously reported for a slightly larger variant of the enzyme. At present we do not understand the reason for this difference. Regardless as suggested by the reviewer, the revised version of the manuscript provides a statement noting the difference and stating that the reason for it remains unknown

(lines 145-146).

Reviewer comment: Satisfied with the author comment.

Minor comments:

1. For section 2.5 and 2.6, the author should use a separate figure using a flowchart and arrow system to describe their High Throughput Screen for inhibitors. This will help the reader to understand the screening process a better way.

We appreciate the difficulty a reader might have in following the respective processes for the pilot and larger high throughput screens involving the two distinct enzymes and so, as suggested, we have now provided the two flowcharts in a Supplemental figure (Fig. S3), and renumbered the remaining supplemental figures accordingly.

Reviewer comment: Satisfied with the author comment.

2. The line 238-241 is confusing or phrased in not a proper way. The author mentioned that the remaining 13 hits showed the IC50 values between 22 and 88uM and referred to figure 4 and S5. But figure 4 is about compound 89224 and S5 is about the false positives. I as a reader found it highly confusing what authors want to point out and which 13 hits they are referring?

We have edited the text to hopefully make more clear the process followed and it outcomes.

Reviewer comment: Satisfied with the author comment.

3. For line 245-250, the author should refer to figure S5 whether they have checked the compounds that interfered with the fluorometric assay. The figure S5 legend should mention which substrate was used.

We have added the citation to (now) figure S6, and added appropriate text to its legend that identifies the substrate used, and meaning of the error bars.

Reviewer comment: Satisfied with the author comment.

4. There are some minor grammatical errors (like the omission of commas at many places) careful proofreading will help.

The manuscript has been checked for grammatical errors. Again, we thank the reviewer for this review.

Reviewer comment: Satisfied with the author comment.

Author Response

Reviewer comment: I think the author should explain it better in the figure legend by incorporating the statement made above to avoid misinterpretation. I agree with the author that N. gonorrhoeae show a sharp decrease in % growth achieved as the concentration of inhibitor is increased. However, for E. coli the second point as compared to the starting point for the % growth achieved also show a significant drop and then become constant. What could be the reason for that drop from the first point to the second point (% growth achieved)? The author should comment on this

We now understand the concern of the reviewer and agree that an additional statement would be helpful.  Thus, the following text has been added to the legend of Figure 5: 

 "The data presented represent the average extents of growth of cultures relative to the respective control cultures. "

 We thank the reviewer for this suggestion.

Reviewer comment: My suggestion was for the better representation of the result shown by the author and interpretation by the readers ( for the overall improvement of the manuscript). As mentioned by the author in line 189-190 “with fluorometric assays, fluorescence quenching by compounds could lead to their false identification as inhibitor”. In my opinion, any compound from the list that does not inhibit the enzyme (negative control) but has the same coumarin core will help in distinguishing the false positives from the actual inhibitor or author can just include the fluorescence curve of inhibitor compound above its inhibitory concentration without enzyme and show that inhibitor does not quench fluorescence by itself above its inhibitory concentration. The negative control as a separate panel in figure 3 will further validate the fluorometric assay used by the author and help in comparing non-inhibitors with the actual inhibitor.

So, if we understand the reviewer’s concern correctly, s/he is requesting that we use a non-inhibitor as some control for potential fluorescence quenching by the inhibitors identified.  But perhaps the reviewer missed the point that we directly tested for any possible fluorescence quenching giving us a false positive result by using the different substrate pNP-Ac for IC50 determinations.  This substrate is not fluorescent but instead chromogenic and so fluorescence quenching is a non-issue. To make this point clear, we have revised the text to include “the chromogenic” pNP-Ac susbstrate on line 194-195 of the revised manuscript.  The fact that we re-tested the inhibitions with a chromogenic assay, rather than a fluorometric one negates the need to attempt the approach suggested by the reviewer. Also, it is likely that the fluorescence properties of the coumarins would be different and so it is not clear to us that a different one would appropriate as a control for others.

Reviewer 2 Report

The authors have addressed the concerns raised.

Author Response

Thank you for your efforts

Reviewer 3 Report

No more comments

Author Response

Thank you for your efforts